# Waste Cooking Oil-Modified Epoxy Asphalt Rubber Binders with Improved Compatibility and Extended Allowable Construction Time

**DOI:** 10.3390/molecules27207061

**Published:** 2022-10-19

**Authors:** Jie Gong, Fan Jing, Ruikang Zhao, Chenxuan Li, Jun Cai, Qingjun Wang, Hongfeng Xie

**Affiliations:** 1MOE Key Laboratory of High Performance Polymer Materials and Technology, School of Chemistry and Chemical Engineering, Nanjing University, Nanjing 210093, China; 2Department of Chemistry, Texas A&M University, College Station, TX 77843-3255, USA; 3Public Instrument Center, School of Chemistry and Chemical Engineering, Nanjing University, Nanjing 210023, China

**Keywords:** crumb rubber, waste cooking oil, epoxy asphalt, viscosity, phase separation, mechanical properties

## Abstract

The application of crumb rubber from end-of-life tires and waste cooking oil (WCO) in road pavements is of significant importance from an economic and environmental viewpoint. However, the incorporation of crumb rubber greatly shortens the allowable construction time of epoxy asphalt binders due to the high viscosity of the epoxy asphalt rubber (EAR) binder and poor compatibility between crumb rubber and asphalt binder. To lower the viscosity of asphalt rubber, extend the allowable construction time and improve the compatibility of EAR binder, waste cooking oil (WCO) was introduced. The effect of WCO on the viscosity–time behavior, thermal stability, dynamic modulus, glass transitions, crosslink density, damping ability, compatibility, mechanical properties and phase separation of WCO-modified EAR binders was investigated by using the Brookfield viscometer, thermogravimetric analysis, dynamic mechanical analysis, universal testing machine and laser confocal microscopy. The test results demonstrated that the incorporation of WCO declined the viscosity and extended the allowable construction time of the unmodified EAR binder. The inclusion of WCO improved the compatibility between asphalt and crumb rubber and the damping ability and elongation at the break of the unmodified EAR binder. The presence of WCO had a marginal effect on the thermal stability of the unmodified EAR binder. Confocal microscopy observation revealed that asphalt rubber particles aggregated in the epoxy phase of the unmodified EAR binder. With the inclusion of WCO, co-continuous asphalt rubber particles became more spherical.

## 1. Introduction

As a thermosetting polymer-modified asphalt, epoxy asphalt is mainly composed of asphalt and epoxy resin. Unlike thermoplastic polymer-modified asphalts, the initial composition of epoxy asphalt includes asphalt and low-molecular-weight, or uncured epoxy resin, which consists of epoxy oligomer and hardener [1,2,3]. To avoid the reaction of the epoxy oligomer to hardener, three components of epoxy asphalt are stored separately. An example of the three-component epoxy asphalt is the hot-mix epoxy asphalt binder. More often than not, asphalt is premixed with hardener. In this case, epoxy asphalt, such as warm-mix and cold-mix epoxy asphalt binders, contains two components: asphalt premixed with hardener and epoxy oligomer. When three components are blended, epoxy oligomer reacts with hardener, which leads to the crosslink of epoxy resin and the incompatibility between cured epoxy and asphalt [4,5,6]. As a consequence, phase-separated morphology forms, in which the continuous phase is determined by the ratio of asphalt to epoxy resin [7,8,9]. Crosslinked networks of cured epoxy completely change the thermoplastic feature of asphalt and confer high strength, excellent fatigue resistance, as well as strong adhesion to aggregates, good temperature and water stability. Therefore, epoxy asphalt materials have been extensively utilized as bond coats and binders in the paving of the steel bridge [10,11,12]. However, the short allowable construction time increases the paving difficulty of epoxy asphalt, especially for warm-mix epoxy asphalt binder (WEAB). Thus, WEAB is seldom used in the pavement rehabilitation and maintenance of steel bridges. To solve this drawback, warm-mix asphalt additives, such as Sasobit and waste cooking oil (WCO), have been introduced into warm-mix epoxy asphalt binder [13,14,15].

Waste cooking oil, mainly composed of triglycerides, is the used vegetable oil that has been heated at a high temperature and collected from food industries, restaurants, hotels and households. It is estimated that the global generation of waste cooking oil is over 16.5 million tons annually [16]. A total of 5.6 million tons of waste cooking oil are available in China every year [17]. As a hazardous and challenging waste, the inappropriate disposal of waste cooking oil in landfills and drains causes soil and water pollution, oxygen scarcity and even the death of flora and fauna in aquatics [18]. In addition, poisonous WCO-born compounds absorbed by aquatic organisms can return to the human food chain [19]. Using waste cooking oil in food industries can give rise to serious diseases, such as stomach aches, dyspepsia and intestinal or gastric cancer [20]. In this case, waste cooking oil also threatens human health. Therefore, more and more attention has been received to the sustainable utilization of waste cooking oil [21]. Among all these strategies, the potential modification of asphalt with waste cooking oil as warm-mix asphalt additives and rejuvenating agents as well as raw materials to produce bio-asphalts has attracted more and more attention [22,23]. For instance, adding WCO declines the relative asphaltene content and increases the light components of asphalt [24]. However, the elastic recovery, high-temperature performance, rutting and deformation resistance of asphalt decrease with the addition of WCO [22]. For epoxy asphalt, the inclusion of WCO not only extends the allowable construction time but also lowers the glass transition temperature (*T*_g_) of the WEAB [15]. However, phase inversion occurs when adding 6% WCO, where asphalt becomes the continuous phase, which highly declines the tensile strength of the WEAB.

On the other hand, fatigue crack after long service periods is one of the main distresses of epoxy asphalt [25]. Without timely treatments, fatigue cracks propagate rapidly under the impact of water and develop into potholes, which severely threaten road safety and shorten the lifetime of pavements [26]. To solve this drawback, inorganic additives, such as basalt and glass fibers [27,28], and polymers, such as styrene-butadiene-styrene copolymer (SBS) [29], epoxidized SBS (ESBS) [30], ethylene vinyl copolymer (EVA) [31], waste polyethylene (PE) [32], core-shell rubber (CSR) [33], hyperbranched polyester (HBP) [34], poly(ethylene glycol) (PEG) [35], polyurethane (PU) [36] and crumb rubber (CR) [37], have been employed to enhance the toughness of epoxy asphalts to relieve or eliminate fatigue cracks. Among all developed modifiers, CR is attracting more and more attention because of its low cost and significant improvement in the asphalt binder’s performance [38]. More importantly, CR used in the production of asphalt rubber binder is a sustainable solution for end-of-life tire disposal, which poses severe environmental pollution and potential health hazards [39]. It is believed that the inclusion of CR improves the high-temperature permanent deformation, low-temperature fatigue crack and noise-reducing capacity of the asphalt binder [40,41,42]. However, CR significantly increases the binder’s viscosity by absorbing the lighter components of the asphalt [43]. For this reason, a higher temperature is needed for the production and pavement of asphalt rubber mixtures in comparison to conventional asphalt mixtures, which results in high energy consumption and the release of large amounts of harmful fumes [44]. Further, asphalt rubber binder has poor storage stability because of the high heterogeneity between CR and asphalt and significant difference in density [45], which becomes a severe challenge for the application of asphalt rubber. In the case of CR modified epoxy asphalt binder, also called epoxy asphalt rubber (EAR) binder, the presence of CR significantly increases the viscosity and thus greatly shortens the allowable construction time [37,46]. Moreover, the heterogeneity between vulcanized rubber and asphalt results in the poor storage stability of the EAR binder.

Recently, the combined use of WCO and CR in the asphalt modification has attracted growing attention, since the shortcomings of the asphalt modified by a single modifier can be overcome by combining the advantages of two modifiers [47]. There are two ways to produce WCO-modified asphalt rubber; the first is by mixing WCO with asphalt rubber and the second is by pre-swelling crumb rubber in WCO and then mixing it with asphalt binder [48]. Due to the improvement of the compatibility between CR and the binder through the chemisorption of WCO and CR [48], WCO has been utilized to improve the storage stability and workability of the asphalt rubber binder. On the other hand, WCO significantly reduces the viscosity of the asphalt rubber [49]. However, to our best knowledge, there are few performance investigations on the modification of epoxy asphalt binder with composite WCO/CR.

In this study, WCO was used an additive to lower the viscosity and the compatibility between the CR and asphalt of EAR binders. To obtain this goal, WCO was mixed with the industrial asphalt rubber binder to prepare the composite WCO/CR-modified asphalt binder. Afterward, epoxy resin was added to the composite modified binder. After curing, the composite WCO/CR-modified epoxy asphalt binder was obtained. The viscosity–time behavior, thermal properties, mechanical performance and microstructures of composite WCO/CR-modified epoxy asphalt binders were investigated. It is believed that an epoxy asphalt rubber binder with a long allowable construction time and good compatibility could be prepared.

## 2. Results and Discussion

### 2.1. Viscosity vs. Time Curves

The rotational viscosity-curing time curves of the unmodified and WCO-modified EAR binders at 160 °C are presented in Figure 1. Unlike conventional asphalt rubber binders, the rotational viscosity of the unmodified EAR binder increases in the curing time because of the curing of epoxy [50]. The inclusion of WCO lowers the rotational viscosity of the unmodified EAR binder. Further, the rotational viscosity of the WCO-modified EAR binders decreases in the oil concentration. That is to say, the existence of WCO dilutes the concentration of active epoxide and hydroxyl moieties of epoxy and thus hinders the curing of epoxy.

The time to reach 1 and 3 Pa·s obtained from the rotational viscosity–time curves are the lower limit and upper limit of epoxy asphalt binder, which are often utilized to determine the shortest and longest allowable construction time of epoxy asphalt mixtures [51]. The lower and upper limits of the unmodified EAR binder are 66 and 94 min, respectively. When adding 2, 4 and 6% WCO, the lower limit increases to 82, 96 and 120 min. At the same time, the upper limit increases to 113, 125 and 144 min, respectively. Compared to the unmodified EAR binder, the shortest and longest allowable construction time of the modified binder containing 6% WCO extend by 82% and 50%, respectively.

### 2.2. Thermal Stability

Figure 2 illustrates the thermogravimetric and differential thermogravimetric (DTG) thermograms of WCO and the unmodified and WCO-modified EAR binders. The one-step thermal decomposition of WCO occurs in the temperature range of 300~500 °C, which is attributed to the degradation of triglycerides [52]. However, the thermal decomposition of the unmodified and WCO-modified EAR binders undergoes two steps. The first one in the temperature region of 250–420 °C is attributed to the release of small gaseous molecules of uncured epoxy and asphalt and the degradation of natural rubber of crumb rubber and triglycerides of WCO. The second one, between 420 °C and 510 °C, is correlated to the decomposition of larger asphalt molecules, cured epoxy and styrene-butadiene rubber and additives of crumb rubbers and triglycerides of WCO [37,53].

The thermal parameters of WCO and the unmodified and WCO-modified EAR binders are summarized in Table 1. Although the onset decomposition temperature (*T*_onset_, the temperature at 5% weight loss) and maximum decomposition rate temperature (*T*_max_) are higher than those of the unmodified EAR binder, the incorporation of WCO has a marginal effect on the *T*_onset_ and *T*_max_ at the first and second decomposition steps (*T*_max1_ and *T*_max2_) of the unmodified EAR binder, which indicates that the thermal stability of the unmodified EAR binder is not altered with the inclusion of WCO.

### 2.3. Dynamic Mechanical Properties

Figure 3 depicts the dynamic modulus-temperature curves of the unmodified and WCO-modified EAR binders. As shown in Figure 3a, the inclusion of WCO lowers the storage modulus (*E′*) of the unmodified EAR binder during the whole temperature region. For WCO-modified EAR binders, the *E′* increased in the WCO concentration in the glassy state, whereas an opposite trend exhibits in the glass transition and rubbery state. Apart from the concentrations of 4% and 6% in the glassy state, the effect of WCO on the loss modulus (*E″*) of the unmodified EAR binder follows the same trend as the *E′*, as shown in Figure 3b.

Figure 4 illustrates the damping factor (tan δ) as a function of the temperature of the unmodified and WCO-modified EAR binders. With the addition of WCO, the main peak (around 30 °C) of the damping factor-temperature curve shifts to a lower temperature. It is known that the peak in a tan δ versus temperature curve typically represents the glass transition of polymeric material [54,55]. Apart from the main peak, another weak peak appears at −30–0 °C in the damping factor-temperature curve. The appearance of two glass transitions of epoxy asphalt is attributed to the inhomogeneity of cured epoxy and asphalt, which results in the occurrence of phase separation [25,56]. For this reason, the higher temperature peak is the *T*_g_ of epoxy resin, while the lower temperature peak is the *T*_g_ of asphalt rubber. Table 2 lists the *T*_g_s of epoxy and asphalt rubber for the unmodified and WCO-modified EAR binders. Apparently, the *T*_g_s of both asphalt rubber and epoxy of the unmodified EAR binder are lowered by the inclusion of WCO. For WCO-modified EAR binders, the *T*_g_s of both the asphalt rubber and epoxy decrease in the oil concentration. WCO-modified warm-mix epoxy asphalt binders and a warm-mix asphalt additive and Sasobit-modified EAR binders showed a similar trend [15,46]. It is believed that the low-temperature properties of asphalt mixtures relate to the *T*_g_ of the asphalt binder [57]. Therefore, the low-temperature performance of the EAR mixture is improved with the inclusion of WCO. In addition, the enhancement effect increases with the WCO concentration.

The *T*_g_ of thermosetting polymers, to a great extent, relies on the crosslink density (*CD*), which is determined by the storage modulus (*E′*_r_) and temperature (*T*_r_) at the rubbery state [58,59]:(1)$$CD=\frac{{E^{\prime }}_{r}}{3RT_{r}}$$
where *R* is the gas constant. *T*_r_ = *T*_g_ + 40 K. As shown in Table 2, the incorporation of WCO significantly declines the crosslink density of the unmodified EAR binder. Further, the crosslink density of WCO-modified EAR binders decreases with the oil concentration. Consequently, WCO lowers the *T*_g_ of the unmodified EAR binder and the *T*_g_ of WCO-modified EAR binders decreases with the oil concentration.

Like epoxy asphalts, the unmodified EAR binder exhibits an outstanding damping ability [46]. Dynamic mechanical analysis (DMA) was used to evaluate the influence of WCO on the damping ability of the unmodified EAR binder. The damping parameters of the unmodified and WCO-modified EAR binders are listed in Table 3 The incorporation of WCO has little effect on the (tan δ)_max_ (maximum damping factor) value of the unmodified EAR binder. However, the Δ*T* (effective damping region where damping factor is over 0.3) of the unmodified EAR binder is widened and *TA* (integral of damping factor vs. temperature curve) is increased with the inclusion of WCO. Especially for 6% WCO, the Δ*T* and *TA* values of the unmodified EAR binder are increased by 9.0 and 8.9 K, respectively. The above outcomes indicate that the damping behavior of the unmodified EAR binder is enhanced with the incorporation of WCO.

Figure 5 shows the loss modulus versus storage modulus curves (also called Cole-Cole plots) of the unmodified and WCO-modified EAR binders. Cole-Cole plots of polymeric materials indicate the compatibility and homogeneity between individual components [60,61]. Due to the inhomogeneity between asphalt and cured epoxy, epoxy asphalt binder exhibits two smooth semicircular curves: the one at high moduli is attributed to cured epoxy and the one at low moduli is related to asphalt [15]. Similar to epoxy asphalt binder, the Cole-Cole plot of the unmodified EAR binder shows two semicircular curves. However, the one at low moduli of asphalt rubber is irregular due to the incompatibility between asphalt and vulcanized rubber. With the inclusion of WCO, the semicircular curves of both asphalt rubber and cured epoxy become smoother. When adding 6% WCO, the semicircular curve of asphalt rubber is nearly as smooth as that of the cured epoxy. These outcomes indicate that the compatibility between asphalt rubber and cured epoxy, particularly for the compatibility between asphalt and vulcanized rubber, is improved with the existence of WCO.

### 2.4. Mechanical Performance

The tensile properties of the unmodified and WCO-modified EAR binders are illustrated in Figure 6. The tensile strength of the unmodified EAR binder (4.87 MPa) is lowered with the inclusion of WCO, because WCO decreases the crosslink density of epoxy resin. However, the tensile strength of the unmodified EAR binder is slightly improved with the inclusion of 1–3% solid Sasobit [46]. The tensile strength of WCO-modified EAR binders decreases in the oil concentration because of the lower crosslink density of epoxy. According to JTG/E3364-02 [62], the tensile strength of hot-mix epoxy asphalt binder is required to be greater than 2 MPa, and all WCO-modified EAR binders meet this specification.

Contrary to the tensile strength, the elongation at the break of the unmodified EAR binder (187%) is enhanced with the inclusion of WCO because the waste oil lowers the crosslink density of the cured binder. The modified EAR binder with 1% Sasobit exhibits a similar improvement [46]. Therefore, the elongation at break of all WCO-modified EAR binders is greater than 100%, which satisfies the specification of JTG/E3364-02 [62]. For WCO-modified EAR binders, the elongation at the break increases with the WCO concentration at first. A maximum value (334%) appears at the concentration of 4%, which is 79 higher than that of the unmodified EAR binder. As the WCO concentration increases, the elongation at the break slightly decreases. The elongation at break of the modified EAR binder with 6% WCO is 71% higher than that of the unmodified EAR binder.

The tensile toughness of the unmodified and WCO-modified EAR binders is depicted in Figure 7. Like the tensile strength, the tensile toughness of the unmodified EAR binder is lowered with the inclusion of WCO. Meanwhile, the tensile toughness of WCO-modified EAR binders decreases in the oil concentration.

### 2.5. Phase-Separated Morphology

Figure 8 illustrates the fluorescence laser confocal microscopy images of the unmodified and WCO-modified EAR binders. Phase-separated microstructures are observed in the unmodified and WCO-modified EAR binders. The yellow continuous phase is cured epoxy since the thermosetting polymer is excited by laser beam and emits strong fluorescence with a longer wavelength [25], whereas the black dispersed phase is asphalt rubber. In comparison to the unmodified EAR binder, the shape of the dispersed phase is more spherical and the dispersion of asphalt rubber particles is more uniform, as shown in Figure. 8b-d. Generally, double phase separation often occurs in polymer-modified epoxy asphalt due to the incompatibility among polymer, asphalt and cured epoxy resin [25]. Furthermore, the added polymer can be dispersed in either the continuous or dispersed phase, which depends on the compatibility between polymer and asphalt or epoxy resin. Due to its good compatibility with asphalt, rubber particles are dispersed in the asphalt, rather than the epoxy. However, different from SBS- and EVA-modified epoxy asphalt binders [31,63,64], it is hard to use the fluorescence confocal microscopy to observe rubber particles in the asphalt phase.

It is important to note that crumb rubber obtained from end-of-life tires contains 45–47% vulcanized rubber [65], which absorbs lighter asphalt components and thus swells in it. More importantly, the laser beam can excite swollen vulcanized rubber and emit fluorescence such as cured epoxy resin. In this case, unlike thermoplastic polymers, such as SBS and EVA, it is challenging to distinguish cured epoxy and vulcanized rubber from the fluorescence confocal microscopy images due to their similar infinite molecular weight networks. Fortunately, laser confocal microscopes collect confocal fluorescence and nonconfocal transmitted light simultaneously by using two detectors that collect fluorescent light emitted by the sample through a confocal pinhole and the light passing through the sample from the same scanning beam [66]. Figure 9 illustrates transmission confocal microscopy images of the unmodified and WCO-modified EAR binders. Surprisingly, the separated asphalt rubber particles in the fluorescence confocal microscopy image (Figure 8a) become aggregates (co-continuous particles) along with several spherical particles of asphalt rubber in the transmission confocal microscopy image (Figure 9a). With the inclusion of 2% WCO, the degree of aggregation decreases and more spherical asphalt rubber particles form (Figure 9b). With the further increase in WCO concentration, nearly all asphalt rubber co-continuous particles of the unmodified EAR binder turn to be spherical (Figure 9c,d). The aggregation of asphalt rubber particles is attributed to the high viscosity of the unmodified EAR binder, which restricts asphalt rubber from forming individual particles during epoxy curing. However, the inclusion of WCO declines the unmodified EAR binder’s viscosity and improves the compatibility between vulcanized rubber and asphalt, which decreases and eventually eliminates the aggregation of asphalt rubber particles.

It needs to be mentioned that asphalt- or polymer-modified asphalt in epoxy asphalt or epoxy polymer-modified asphalt rearranges during the formation of phase-separated microstructures [31,67,68,69]. Figure 10 presents transmission confocal microscopy images of the unmodified asphalt rubber binder and the modified one with 12% WCO. Unfortunately, the phase-separated microstructures of rubber and asphalt are invisible, which may be due to the existence of large amounts of additives, such as carbon black [65]. Instead, crumb rubber particles are observed in the confocal microscopy image of the unmodified asphalt rubber binder (Figure 10a). With the incorporation of WCO, crumb rubber particles disappear and aggregates of carbon black are observed due to the dilute and compatible effect of the waste oil (Figure 10b). For the unmodified EAR binder, the existence of 50% epoxy resin significantly lowers the viscosity of the unmodified asphalt rubber binder. In this case, co-continuous particles caused by the aggregation of asphalt rubber, accompanied by several spherical particles, form during the occurrence of phase separation. The inclusion of WCO further dilutes the concentration of the unmodified asphalt rubber binder and improves the compatibility between asphalt and vulcanized rubber and between asphalt rubber and cured epoxy resin as proved by Cole-Cole plots. In this case, co-continuous asphalt rubber particles become spherical ones, which are dispersed more uniformly in the cured epoxy phase.

The area fraction, average diameters and dispersity of asphalt rubber particles of the unmodified and WCO-modified EAR binders are depicted in Table 4. The inclusion of 2% WCO increases the number-average diameter (*D*_n_) of the unmodified EAR binder, while a contrary trend is shown in the 6% WCO. The incorporation of WCO decreases the weight-average diameter (*D*_w_) of the unmodified EAR binder. Besides, the *D*_w_ of WCO modified EAR binders decreases in the oil concentration. The existence of WCO greatly decreases the polydispersity index (*PDI*, the ratio of *D*_w_ to *D*_n_). For WCO-modified EAR binders, the *PDI* value decreases in the oil concentration. The above outcomes reveal that asphalt rubber particles are dispersed more uniformly in the cured epoxy phase with the existence of WCO. As proved by Cole-Cole plots, the compatibility between asphalt and crumb rubber increases with the WCO concentration. Thus, the dispersion of asphalt rubber particles becomes more uniform, as shown in Figure 9b–d.

The 2% WCO has little effect on the area fraction of asphalt rubber particles of the unmodified EAR binder, as illustrated in Table 4. However, adding 4% and 6% WCO lowers the area fraction of the dispersed phase. At the same time, the modified EAR binder with 4% WCO has a lower area fraction than the binder with 6% WCO. It was reported that both crumb rubber and asphalt binder interact with WCO, resulting in the swelling of vulcanized rubber and the increase in light components of the binder [70]. Due to the rubber absorption of both the oil and light components of asphalt, the area fraction of rubber particles in the cure epoxy phase decreases when adding more than 4% WCO. Besides, a balance of the rubber absorption levels out between 4% and 6% WCO. Therefore, the area fraction of the dispersed phase for the modified EAR binder with 6% WCO increases. Noteworthily, due to the rubber absorption, the phase-separated structure of the modified EAR binder containing 6% WCO remains. Nevertheless, the phase inversion takes place in the epoxy asphalt binder with the same WCO concentration because of the area fraction increase in asphalt [15]. In other words, asphalt becomes the continuous phase. The epoxy asphalt containing 60% SBS-modified asphalt demonstrates a similar phenomenon [63].

## 3. Materials and Methods

### 3.1. Materials

The industrial asphalt rubber binder containing 20% CR by weight and Pen 60/80 asphalt binder were obtained from China Offshore Bitumen (Taizhou) Co. Ltd. (Taizhou, China). The physical properties of the asphalt rubber binder are summarized in Table 5. The WCO collected from household kitchens was waste peanut oil. Table 6 lists the physical property and chemical constitution of WCO. Figure 11 shows the asphalt rubber binder and WCO used in this study. The epoxy resin was prepared in the laboratory. Table 7 presents the physical properties of epoxy oligomer and hardener. Figure 12 illustrates the two components of epoxy resin used in this study.

### 3.2. Preparation of Asphalt Rubber Binder

Asphalt rubber binder containing 20% by weight CR was preheated at 190 °C and mixed with pen 60/80 asphalt binder preheated at 150 °C in a beaker using a high-shear mixer at 190 °C and a speed of 4000 min^−1^ for 1 h. The mass ratio of asphalt rubber binder to virgin asphalt binder is 3:1.

### 3.3. Preparation of WCO-Modified Asphalt Rubber Binders

WCO was mixed with asphalt rubber binder containing 15% CR preheated at 160 °C in a beaker using a mechanical mixer at 200 min^−1^ and a temperature of 160 °C for 30 min. The mass fractions of WCO in WCO-modified asphalt rubber binders are 4, 8 and 12%, respectively.

### 3.4. Preparation of WCO-Modified EAR Binders

WCO-modified asphalt rubber binder was mixed with the hardener in a beaker at 160 °C at 200 min^−1^ for 30 min. Then, epoxy oligomer was introduced and mixed at 200 min^−1^ for 5 min. Finally, the uncured WCO-modified EAR binder was poured into a Teflon mold and cured at 150 °C for three hours and 60 °C for three days. The mass ratio of the WCO-modified asphalt rubber binder, hardener and epoxy oligomer in a WCO-modified EAR binder is 100:51:49. Due to the dilution effect of epoxy resin, the mass fractions of WCO in WCO-modified EAR binders are 2, 4 and 6%, respectively. Figure 13 presents the preparation scheme of the WCO-modified EAR binder.

### 3.5. Methods

The rotational viscosity-curing time behavior was determined using a Changji NDJ-1C Brookfield rotational viscometer (Shanghai, China). The measurement was conducted with the spindle of 28 at 160 °C. Rotational viscosity was recorded every 5 min until it reached 5000 mPa·s.

Thermogravimetric analysis (TGA) was conducted on a Mettler Toledo TGA/DSC1 system (Zurich, Switzerland). The sample was heated from 50 °C to 600 °C at 20 °C min^−1^ under the protection of nitrogen.

Dynamic mechanical properties were evaluated using a 01 dB-Metravib DMA + 450 dynamic mechanical analyzer (Limonest, France) with a tension mode. Cuboid samples (20 × 20 × 3 mm^3^) were heated from −50 to 100 °C at a ramp rate of 3 °C min^−1^ and 1 Hz.

Tensile properties were determined using an Instron 4466 universal testing machine (Norwood, MA, USA) with a 10 kN load cell. Five duplicates for every sample were tested at a speed of 200 mm min^−1^ at room temperature according to ASTM D 638.

Phase-separated microstructures were observed using a Zeiss LSM710 laser confocal microscope (Jena, Germany). The slide sample was prepared as follows: the mixture of WCO-modified EAR binder was dissolved in toluene for 24 h. A drop of 2 g mL^−1^ solution on a microscope slide was spin-coated at 3000 min^−1^ for 60 s. Then, the solvent of the slide sample was removed at 115 °C for 3 min before covering a cover slide. At last, the sample slide was cured at 150 °C for 3 h followed by 60 °C for 3 days. The sample slide of the WCO-modified asphalt rubber was prepared by the same procedure except for the curing step. The area fraction and average diameters of the dispersed phases of transmission confocal microscopy images with the magnification of ×100 were determined by Image-Pro Plus software.

## 4. Conclusions

This paper investigates the impact of waste cooking oil on the properties and phase separation of epoxy asphalt rubber binders. The incorporation of WCO declines the viscosity and extends the time to both lower and upper viscosity limits of the unmodified EAR binder. The viscosity and allowable construction time of the WCO-modified EAR binders decrease in the oil concentration. With the inclusion of 6% WCO, the lower and upper viscosity limits of the unmodified EAR binder increase by 82% and 50%, respectively. The inclusion of WCO lowers the dynamic modulus of EAR at the rubbery region and the *T*_g_s of both epoxy and asphalt of the unmodified EAR binder. However, the low-temperature properties and damping ability of the unmodified EAR binder and the compatibility between asphalt and crumb rubber were improved with the addition of WCO. Furthermore, the improvement effect increases in the WCO concentration. The existence of WCO improves the elongation at the break of the unmodified EAR binder. The mechanical properties of all modified EAR binders containing less than 6% WCO meet the specification of hot-mix epoxy asphalt binders used on a highway steel bridge. Phase separation results in the aggregation of asphalt rubber in the cured epoxy phase. The co-continuous asphalt rubber particles in the unmodified EAR binder turn to be uniformly spherical with the inclusion of WCO. Overall, this study has successfully extended the allowable construction time and improved the compatibility between asphalt rubber and cured epoxy with the incorporation of WCO. Further studies on the performance of the WCO-modified EAR mixture are left to be conducted. The combined use of these two wastes could help to alleviate the environmental pollution of these disposals.

## Figures and Tables

**Figure 1 molecules-27-07061-f001:**
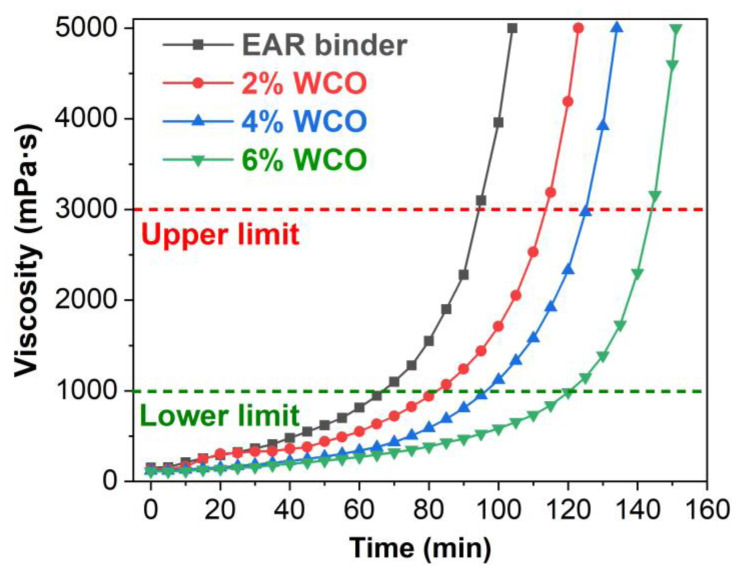
Rotational viscosity as a function of curing time of the unmodified and WCO-modified EAR binders at 160 °C.

**Figure 2 molecules-27-07061-f002:**
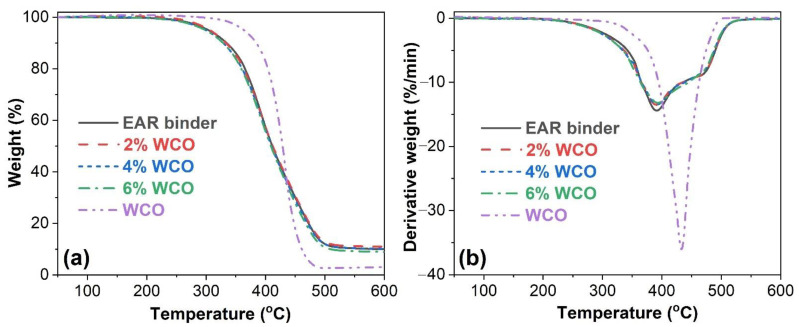
(**a**) TGA and (**b**) DTG curves of WCO and the unmodified and WCO-modified EAR binders.

**Figure 3 molecules-27-07061-f003:**
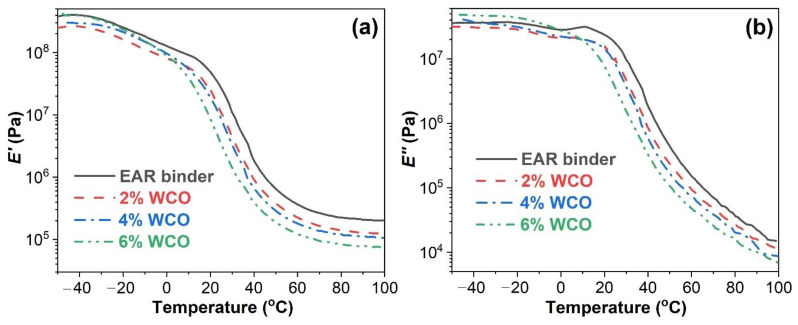
(**a**) *E′* and (**b**) *E″* vs. temperature of the unmodified and WCO-modified EAR binders.

**Figure 4 molecules-27-07061-f004:**
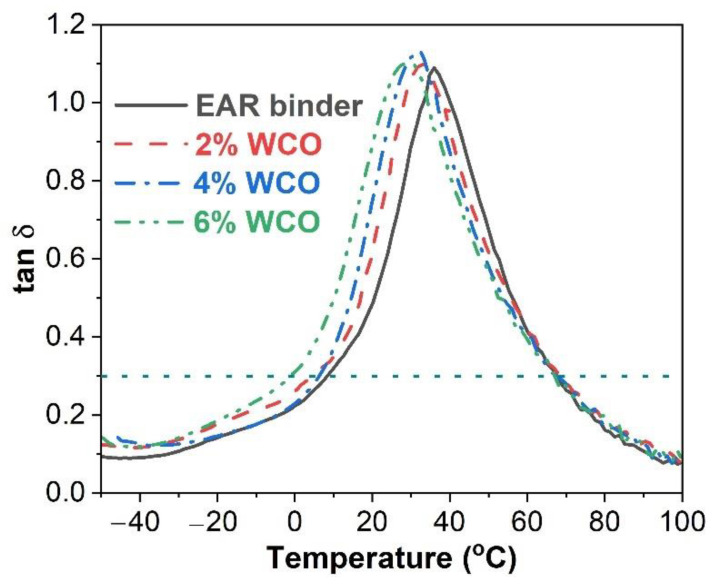
tan δ vs. temperature curves of the unmodified and WCO-modified EAR binders.

**Figure 5 molecules-27-07061-f005:**
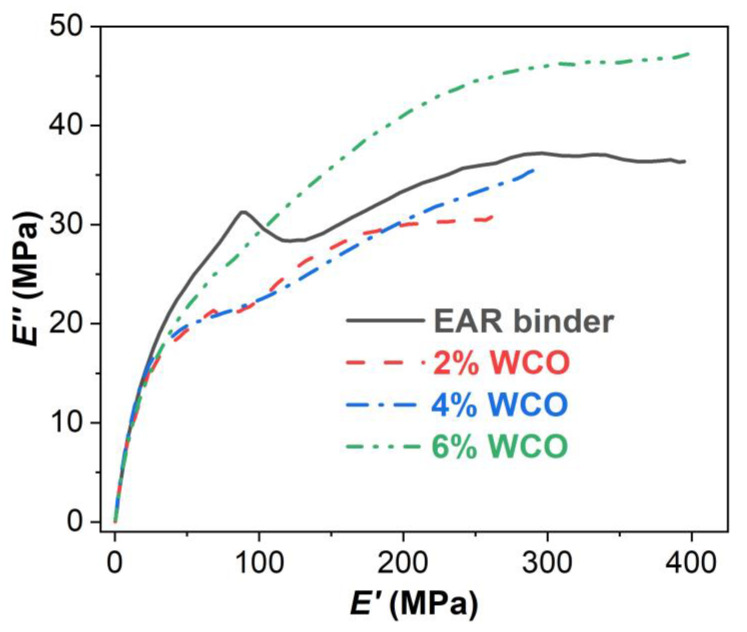
Cole-Cole plots of the unmodified and WCO-modified EAR binders.

**Figure 6 molecules-27-07061-f006:**
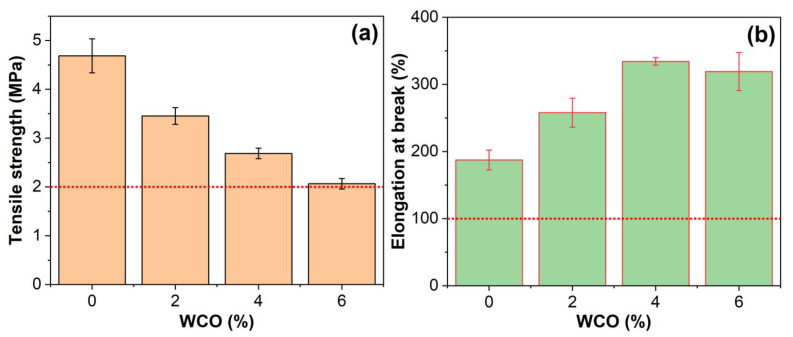
(**a**) Tensile strength and (**b**) elongation at break of the unmodified and WCO-modified EAR binders.

**Figure 7 molecules-27-07061-f007:**
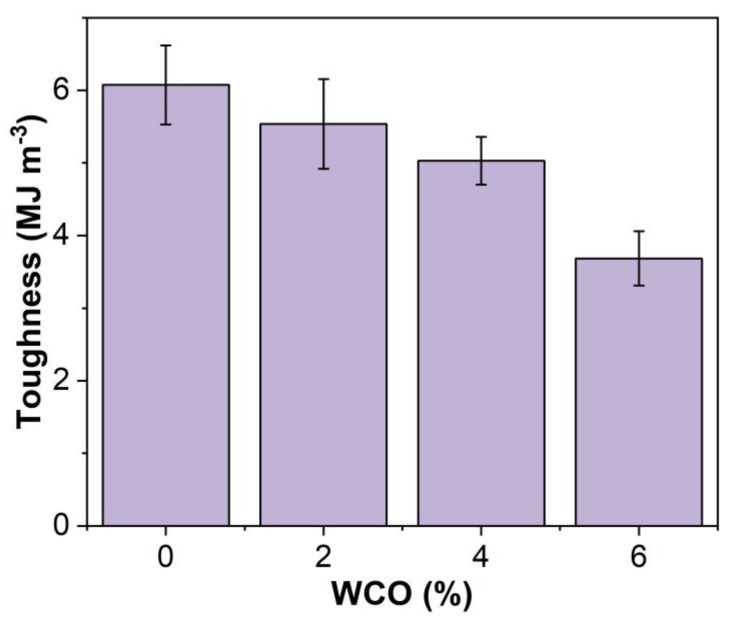
Tensile toughness of the unmodified and WCO-modified EAR binders.

**Figure 8 molecules-27-07061-f008:**
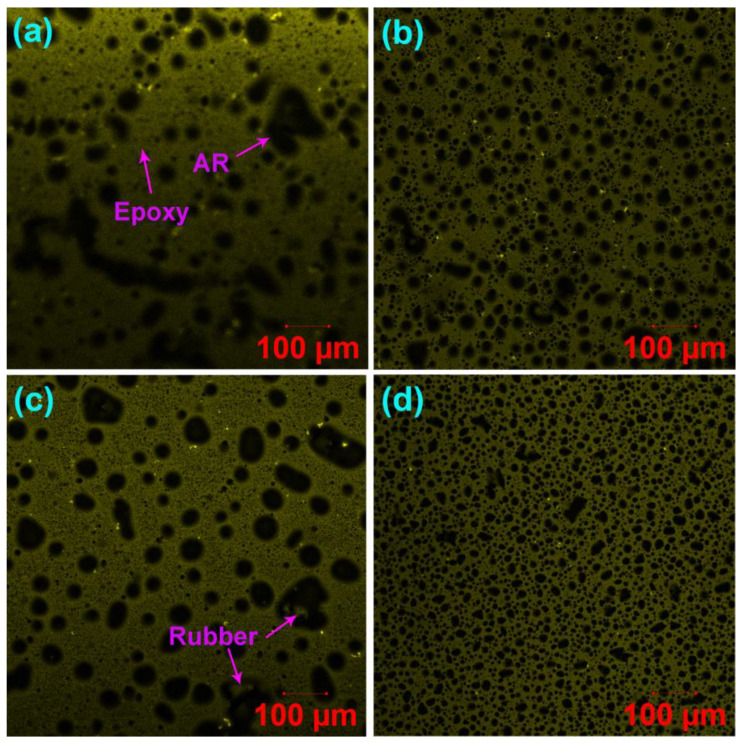
Fluorescence laser confocal microscopy images of (**a**) the unmodified EAR binder and modified EAR binders containing different WCO concentrations: (**b**) 2%, (**c**) 4% and (**d**) 6%.

**Figure 9 molecules-27-07061-f009:**
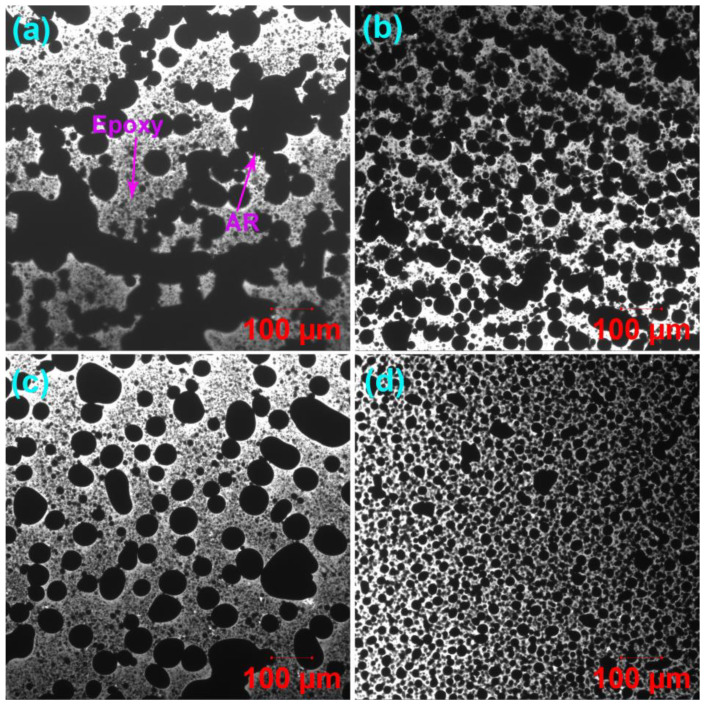
Transmission laser confocal microscopy images of (**a**) the unmodified EAR binder and modified EAR binders containing different WCO concentrations: (**b**) 2%, (**c**) 4% and (**d**) 6%.

**Figure 10 molecules-27-07061-f010:**
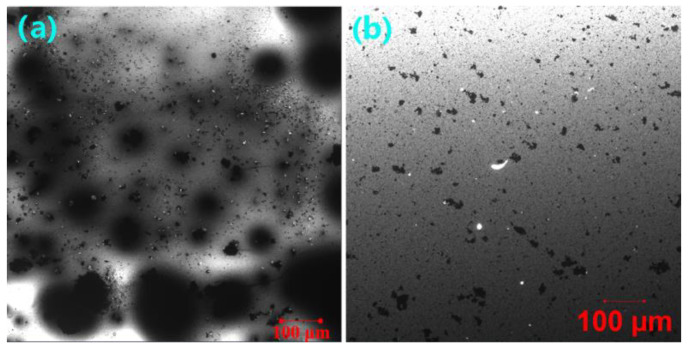
Transmission laser confocal microscopy images of (**a**) the unmodified asphalt rubber binder and (**b**) the modified asphalt rubber binder with 12% WCO.

**Figure 11 molecules-27-07061-f011:**
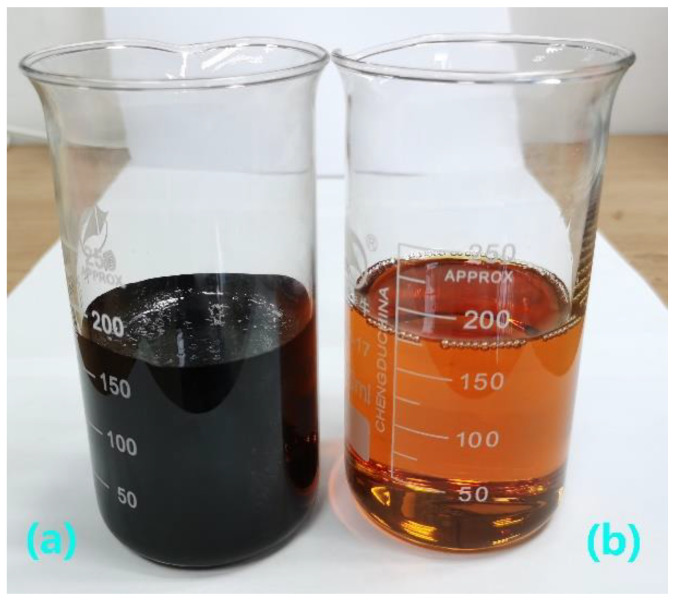
(**a**) Asphalt rubber binder and (**b**) waste cooking oil used in this study.

**Figure 12 molecules-27-07061-f012:**
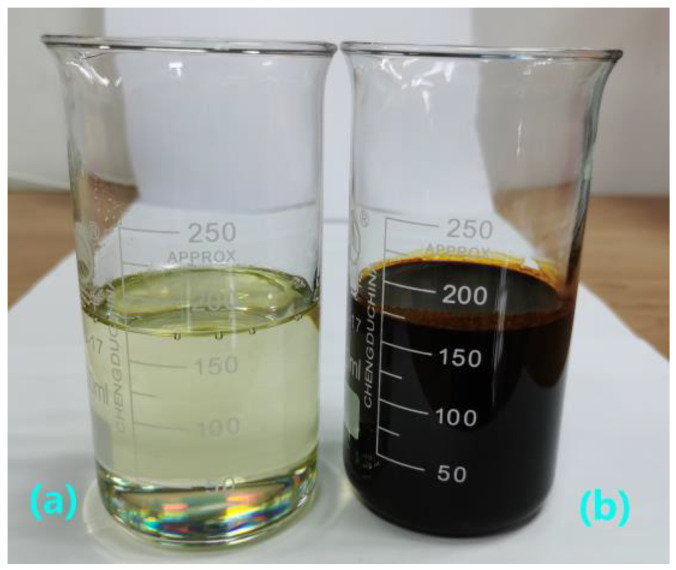
Epoxy resin used in this study: (**a**) epoxy oligomer and (**b**) hardener.

**Figure 13 molecules-27-07061-f013:**
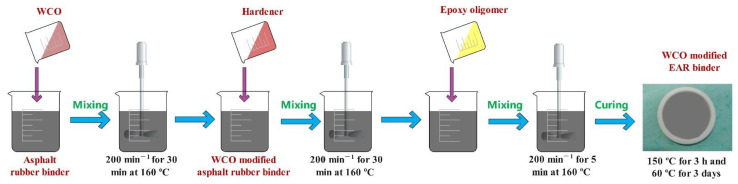
Preparation scheme of WCO-modified EAR binder.

**Table 1 molecules-27-07061-t001:** Thermal parameters of WCO and the unmodified and WCO-modified EAR binders.

WCO (%)	*T*_onset_ (°C)	*T*_max1_ (°C)	*T*_max2_ (°C)
0	304	391	435
2	308	391	436
4	300	392	433
6	300	393	435
100	364	433	-

**Table 2 molecules-27-07061-t002:** Glass transition temperatures and crosslink densities of the unmodified and WCO-modified EAR binders.

WCO (%)	*T*_g_ of Epoxy (°C)	*T*_g_ of Asphalt Rubber (°C)	*CD* (mol m^−3^)
0	35.9	−14.8	27.2
2	33.0	−16.7	18.1
4	31.2	−18.9	15.9
6	28.7	−22.7	15.7

**Table 3 molecules-27-07061-t003:** Damping parameters of the unmodified and WCO-modified EAR binders.

WCO (%)	(tan δ)_max_	Δ*T* (K)	*TA* (K)
0	1.09	59.4 (8.3–67.7)	51.5
2	1.10	65.4 (4.1–69.5)	56.0
4	1.13	61.5 (6.2–67.7)	55.5
6	1.10	68.4 (−1.3–67.1)	59.4

**Table 4 molecules-27-07061-t004:** Average diameters, dispersity and area fraction of dispersed phase of the unmodified and WCO-modified EAR binders.

WCO (%)	*D*_n_ (μm)	*D*_w_ (μm)	*PDI*	Area Fraction of Dispersed Phase (%)
0	32.8	210.4	6.41	58.4
2	37.8	128.2	3.40	59.3
4	32.9	62.7	1.91	42.5
6	24.5	38.1	1.56	47.9

**Table 5 molecules-27-07061-t005:** Physical property of industrial asphalt rubber binder.

Property	Asphalt Rubber Binder
Penetration (25 °C, 0.1 mm)	50
Softening point (°C)	68.0
Viscosity (170 °C, Pa·s)	4.0

**Table 6 molecules-27-07061-t006:** Physical property and chemical constitution of WCO.

Property	Asphalt Rubber Binder
Color	Light brown
Density (25 °C, g cm^−3^)	0.925
Viscosity (25 °C, mPa·s)	135
Acid value (mg KOH/g)	2.36
Iodine value (g I/g)	99
Saponification value (mg KOH/g)	190
Moisture content (%)	0.12
Palmitic acid (%)	3.6
Stearic acid (%)	20.2
Linoleic acid (%)	61.8
Oleic acid (%)	2.7

**Table 7 molecules-27-07061-t007:** Physical property of the epoxy resin for preparing epoxy asphalt binder.

Property	Epoxy Oligomer	Hardener
Viscosity (25 °C, mPa·s)	5800	65
Density (25 °C, g cm^−^^3^)	1.16	0.95
Color	Yellow liquid	Brown liquid

## Data Availability

All data are available in the manuscript.

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
