# Peer review of "Waste Cooking Oil-Modified Epoxy Asphalt Rubber Binders with Improved Compatibility and Extended Allowable Construction Time"

_molecules, 2022, doi:10.3390/molecules27207061_

Round 1
Reviewer 1 Report
The manuscript introduces a novel chemical liquid mix that incorporates recycled tire waste rubber oil and wastes cooking oil in the asphalt binder to produce a durable recycled asphalt bonding by overcoming the overall viscosity and compatibility challenges between the crumb rubber and asphalt binder.
The experimental concept, process, diagrams, tables, graphs, and results are clearly presented. Also, the exploratory outcome discussions are briefly explained.
Author Response
Thank you.
Reviewer 2 Report
This study prepared waste cooking oil (WCO) modified epoxy asphalt rubber binders and evaluated its viscosity-time behavior, thermal stability, dynamic modulus, glass transition, mechanical properties and microstructure. It has been well written and well-oriented. There are some comments:
1) The Abstract should be enriched with the brief details of the experimental methodology. The problem to be addressed in this study should also be highlighted in the Abstract.
2) The novelty and significance of the present work should be highlighted in the last paragraph of the Introduction section.
3) The following related researches could be reviewed:
Laboratory Investigation of The Recycled Asphalt Concrete with Stable Crumb Rubber Asphalt Binder. Construction and Building Materials, 2019, 203: 552-557. doi: 10.1016/j.conbuildmat.2019.01.114
Property Characterization of Asphalt Binders and Mixtures Modified by Different Crumb Rubbers, Journal of Materials in Civil Engineering, 2017, 29(7): 04017036-1-10. doi: 10.1061/(ASCE)MT.1943-5533.0001890
4) Line 90: please check “low-temperature fatigue cracking”;
5) Revise unit of temperature;
6) Why did the authors prepare asphalt rubber binder with 15% by mixing the binder containing 20% CR with virgin asphalt binder? It might be possible to directly utilize the binder with 20% CR.
7) How did the author define the onset decomposition temperature?
8) The authors state the addition of WCO enhanced low-temperature performance of EAR mixture due to the decreased glass transition temperature. Please confirm the relationship between glass transition temperature and low-temperature performance.
9) Line 259: what is the rubbery state? Is Eq.(1) appropriate to be applied in WCO modified EAR binders?
10) Line 301: it is suggested to delete this sentence. The inclusion of Sasobit is not related to this work.
11) The authors state that both epoxy and vulcanized rubber emit fluorescence. How different is fluorescent intensity between both phases?
12) How did the authors derive dispersity, Dn and Dw of asphalt rubber particles in modified EAR binders? It is suggested to report area fraction and average diameter based on three and more confocal microscopy images.
Author Response
Thank you. The labelling for Figures 11 and 12 has been revised.